# ⊛Cuff-KT: Tackling Learners' Real-time Learning Pattern Adjustment via Tuning-Free Knowledge State-Guided Model Updating

## Abstract

Knowledge Tracing (KT) is a core component of Intelligent Tutoring Systems, modeling learners' knowledge state to predict future performance and provide personalized learning support. Current KT models simply assume that training data and test data follow the same distribution. However, this is challenged by the continuous changes in learners' patterns. In reality, learners' patterns change irregularly at different stages (*e.g.*, different semesters) due to factors like cognitive fatigue and external stress. Additionally, there are significant differences in the patterns of learners from various groups (*e.g.*, different classes), influenced by social cognition, resource optimization, etc. We refer to these distribution changes at different stages and from different groups as intra-learner shift and inter-learner shift, respectively—a task introduced, which we refer to as Real-time Learning Pattern Adjustment (RLPA). Existing KT models, when faced with RLPA, lack sufficient adaptability, because they fail to timely account for the dynamic nature of different learners' evolving learning patterns. Current strategies for enhancing adaptability rely on retraining, which leads to significant overfitting and high time cost problem. To address this, we propose Cuff-KT, comprising a controller and a generator. The controller assigns value scores to learners, while the generator generates personalized parameters for selected learners. Cuff-KT adapts to distribution changes fast and flexibly without fine-tuning. Experiments on one classic and two latest datasets demonstrate that Cuff-KT significantly improves current KT models' performance under intra- and inter-learner shifts, with an average relative increase of 7% on AUC, effectively tackling RLPA. [1]

## 1 Introduction

For nearly a century, researchers have been dedicated to developing Intelligent Tutoring Systems (ITS) (Pressey, 1926; Kamalov et al., 2023; Zhou et al., 2024). *Knowledge Tracing (KT), as a core component of ITS, aims to model learners' knowledge state during their interactions with ITS to predict their performance on future questions* (Corbett & Anderson, 1994), as shown in Figure 1. Solving the KT problem can help teachers or systems better identify learners who need further attention and recommend personalized learning materials to them (Liu et al., 2021; Abdelrahman et al., 2023; Liu et al., 2023b).

Reviewing the current research on KT (Piech et al., 2015; Shen et al., 2022; Liu et al., 2023a), we can systematize a dominant paradigm: using learners' historical interaction sequences as training data, encoding them into representations with KT models, and then using these representations to predict future interactions in the test data. This

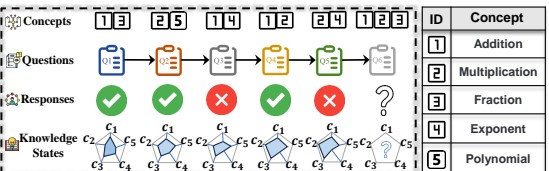

Figure 1: Illustration of the knowledge tracing (KT).

paradigm simply assumes that the training data and test data come from the same distribution. However, this assumption is difficult to hold in real-world scenarios, as it ignores the dynamic nature

---

[1]Our code and datasets are available at `https://anonymous.4open.science/r/Cuff-KT`.

of KT. Specifically, the properties of streaming data (*e.g.*, the correct rate distribution) often change across different stages or groups (Zhang et al., 2017; Yang et al., 2023), indicating that the sequential patterns of learners at different stages or from different groups dynamically vary between historical and future interactions. We refer to these distribution changes across different stages and groups as intra-learner shift and inter-learner shift, respectively.

Distribution shift caused by varying sequential patterns undermines current KT models, resulting in deteriorated generalization when serving future data. Figure 2 provides empirical evidence of this issue. We first divide the assist15 data into 4 non-overlapping parts by stage and group (see Section 4.3 for the division method), and calculate the KL-divergence $w.r.t.$ correct rate distribution between the first part and the other parts. We then train the DKT (Piech et al., 2015) model on the first part and test it on the remaining parts. Clearly, as the KL-divergence increases across different stages or groups, the model's predictive performance significantly declines. Therefore, it is crucial to enhance the dynamic adaptability of KT models. To this end, we introduce a new task, Real-time Learning Pattern Adjustment (RLPA), to address the inability of existing KT models to effectively handle distribution changes arising from differing learning patterns across various stages or groups.

To tackle RLPA, a well-known generalization technique is to retrain (*e.g.*, fine-tune) the pre-trained KT model based on data from the current stage or group to achieve personalized learning (Houlsby et al., 2019; Zaken et al., 2021; Han et al., 2024). Although fine-tuning based approaches are promising, they may not be the optimal solution due to two key

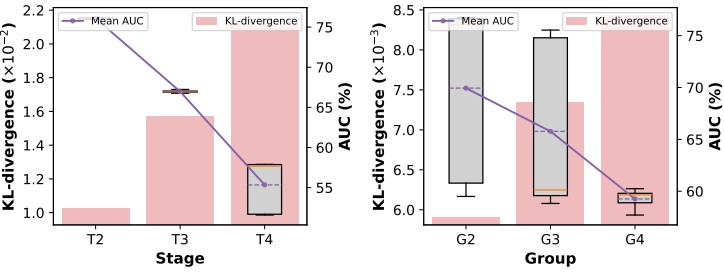

(a) Intra-learner shift      (b) Inter-learner shift

Figure 2: Empirical evidence of model generalization deterioration under different shifts.

challenges: (i) Overfitting. To achieve personalized learning, fine-tuning based approaches often require retraining the model based on very limited samples with rapidly changing distributions, which may lead to overfitting, potentially reducing its ability to generalize to future distributions. (ii) High time cost. Fine-tuning is very time-consuming as it requires extensive gradient computations to update model parameters, which is cumbersome in real-world scenarios where real-time requirements are common. Therefore, fine-tuning based methods must carefully balance the need to adapt to recent data and maintain robustness to achieve generalization. These challenges prompt us to reconsider the design of better solutions to the RLPA in KT.

Towards this end, we propose a novel method to trackle RLPA in KT, called **C**ontrollable, t**U**ning-free, **F**ast, and **F**lexible **K**nowledge **T**racing (**Cuff-KT**). Unlike fine-tuning-based approaches that produce **updated** parameters, the core idea of Cuff-KT is to learn a model parameter generator specific to the current stage or group, generating **updating** personalized parameters for valuable learners (*e.g.*, those showing significant progress or regression), achieving adaptive generalization. Our Cuff-KT consists of two modules: a controller and a generator. **When the data distribution of learners changes due to varying learning patterns, the KT model generalizes worse to the current data and tends to make incorrect evaluations. This implies that the benefit of generating parameters is significant, as generated parameters can appropriately model the current data distribution.** The controller, while considering the fine-grained distance between knowledge state distributions across various concepts, is also inspired by the Dynamic Assessment Theory (Vygotsky & Cole, 1978) and integrates coarse-grained changes in correct rates, assigning a value score to each learner. The generator generates parameters for learners selected based on the assigned value scores[2] by the controller and enhances adaptability. Specifically, considering the relative relationship between question difficulty and learner ability (Rasch, 1993; Shen et al., 2022) and inspired by the dual-tower model in recommendation systems (Huang et al., 2013), the generator models questions and responses separately, extracts features through a sequential feature extractor, simulates the distribution of real-time samples from the current stage or group to achieve adaptive generaliza-

---

[2]The larger a learner's value score, the more likely they are to be selected.

tion through our designed state-adaptive attention, and finally reduces the parameter size through low-rank decomposition. Notably, our generator can be inserted into into any layer or generate parameters for any layer.

Our contributions are summarized as follows:

- We introduce a new task, **RLPA**, which enhances the adaptability of existing KT models in the realm of personalized learning, addressing the challenges arising from distribution shifts caused by varying sequential patterns of learners across different stages or groups.

- We propose **Cuff-KT**, a controllable, tuning-free, fast, and flexible general neural method, which can effectively generate parameters aligned with the current stage or group's learner distribution and insert them into any layer of existing KT models. It is noteworthy that Cuff-KT is model-agnostic.

- We instantiate one classic KT model and two latest state-of-the-art models. Experiments on one classic dataset and two latest datasets demonstrate that our proposed Cuff-KT generally improves current KT models under both intra- and inter-learner shift. Specifically, the AUC metric, which is most commonly used in KT, has relatively increased by 7% on average, proving that Cuff-KT can effectively tackle RLPA in KT.

## 2 RELATED WORK

Knowledge tracing (KT), the task of dynamically modeling a learner's knowledge state over time, traces its origins back to the early 1990s, with early notable contributions by Corbett and Anderson (Corbett & Anderson, 1994). However, with the rise of deep learning, KT research has gained significant momentum, leading to the development of more sophisticated and refined models capable of capturing the intricate dynamics of learner learning (Piech et al., 2015; Yeung & Yeung, 2018; Shen et al., 2022; Liu et al., 2023a). DKT (Piech et al., 2015) first applies LSTM to KT to model the complex learners' cognitive process, bringing a leap in performance compared to previous KT models (*e.g.*, BKT (Corbett & Anderson, 1994)). Subsequently, various neural architectures (*e.g.*, attention and graphs) begin to be introduced into KT (Pandey & Karypis, 2019; Nakagawa et al., 2019; Ghosh et al., 2020; Pandey & Srivastava, 2020). Meanwhile, some training techniques (*e.g.*, adversarial training and contrastive learning) also start to be used in KT research (Guo et al., 2021; Lee et al., 2022a). Recently, incorporating learning-related information has been explored to enhance the predictive capability of KT models. For instance, DIMKT (Shen et al., 2022) improves KT performance by establishing relationship between learners' knowledge states and question difficulty levels, while AT-DKT (Liu et al., 2023a) addresses the issues of sparse representation and personalization in DKT by introducing two auxiliary learning tasks: question tagging prediction and individualized prior knowledge prediction.

However, surprisingly, to our knowledge, there is a lack of attention to adaptability in KT research, which severely affects the generalization of KT models across different distributions. Meanwhile, related studies (Wong et al., 2022; Wong & Ramasamy, 2024) are limited in their applicable scenarios (*e.g.*, continuously increasing learners or concepts) and do not provide a general method to enhance adaptability in KT. Thanks to the well-known fine-tuning based methods, the adaptability in KT has been enhanced to some extent. However, the challenges posed by overfitting and high time cost of fine-tuning based methods make it difficult to be effectively applied in real-world scenarios (Lv et al., 2023b). Even the recently proposed parameter-efficient fine-tuning based methods (*e.g.*, Adapter-based tuning (Houlsby et al., 2019) and Bias-term Fine-tuning (Zaken et al., 2021)) still incur non-negligible time cost and cannot avoid the potential risk of overfitting. Our Cuff-KT, in contrast, updates KT models under dynamic distributions through controllable parameter generation, eliminating the need for retraining and providing a new perspective on enhancing adaptability in the KT community.

## 3 METHODOLOGY

In this section, we first define the problem of KT and formalize the RLPA task in KT, then introduce our proposed Cuff-KT method.

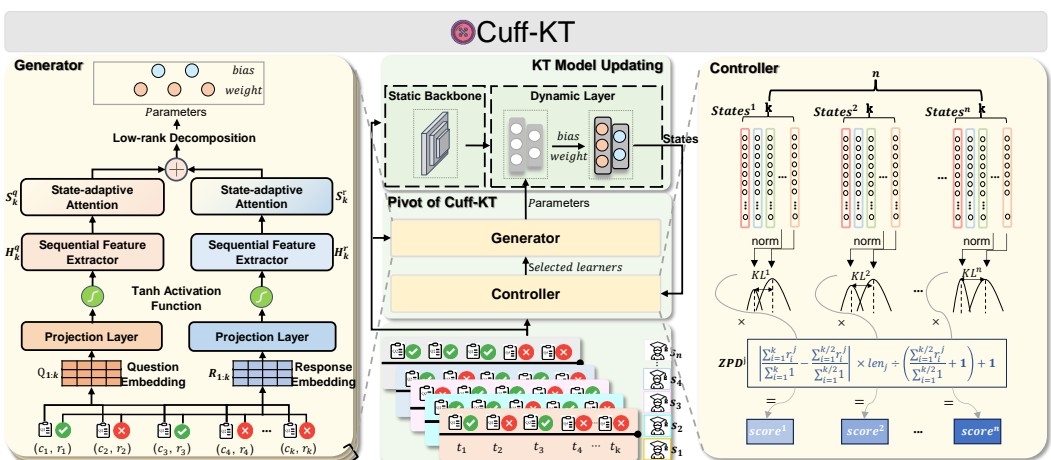

Figure 3: Overview of proposed Cuff-KT method.

## 3.1 PROBLEM FORMULATIONS

### 3.1.1 KNOWLEDGE TRACING

Formally, let $\mathcal{S}$, $\mathcal{Q}$, and $\mathcal{C}$ denote the sets of learners, questions, and concepts, respectively. For each learner $s \in \mathcal{S}$, their interactions are represented $X^s = \{x\}_{i=1}^k$ at time-step $k$, where the interaction $x$ is defined as a 4-tuple, $i.e.$, $x = (q, \{c\}, r, t)$, where $q \in \mathcal{Q}, \{c\} \subset \mathcal{C}, r, t$ represent the question attempted by the learner $s$, the concepts associated with the question $q$, the binary variable indicating whether the learner responds to the question correctly (1 for correct, 0 for incorrect), and the timestamp of the learner's response respectively. The goal of KT is to predict $\hat{r}_{k+1}$ given the learner's historical interactions $X$ and the current question $q_{k+1}$ at time-step $k + 1$.

### 3.1.2 RLPA TASK

RLPA aims to address two common shift issues (intra- and inter- shifts) in KT to enhance the adaptability of existing models. An interaction sequence of a learner $s$ can be divided into multiple stages, assuming each stage has a length of $L$. At time-step $u$, the representation of the learner's interaction in that stage is $X_u^s = X_{u:u+L-1}^s$. Intra-learner shift is defined as: for any time-step $u \neq v$,

$$|d(\chi_u^s, \chi_v^s)| > \delta, \tag{1}$$

where $\delta$ is a small threshold. $\chi_u^s$ and $\chi_v^s$ represent the distributions of $X_u^s$ and $X_v^s$ respectively. $d$ is a distance function ($e.g.$, KL divergence). In contrast, inter-learner shift is:

$$|d(\chi_u^s, \chi_u^{s^*})| > \delta, \tag{2}$$

where $\chi_u^s$ and $\chi_u^{s^*}$ represent the distributions of learners $s$ and $s^*$ at time-step $u$, respectively.

When equations 1 or 2 hold, the goal of RLPA is to adjust the parameters of the existing KT model in real-time so that the predicted distribution $\hat{\chi}_v^s$ or $\hat{\chi}_u^{s^*}$ is as close as possible to the actual distribution:

$$\min_{\hat{\chi}_v^s} \sum_x \chi_v^s(x) \log(\frac{\chi_v^s(x)}{\hat{\chi}_v^s(x)}) \text{ or } \min_{\hat{\chi}_u^{s^*}} \sum_x \chi_u^{s^*}(x) \log(\frac{\chi_u^{s^*}(x)}{\hat{\chi}_u^{s^*}(x)}), \tag{3}$$

where $x$ is a variable in the sample space.

## 3.2 CUFF-KT

Figure 3 illustrates an overview of our Cuff-KT method, which consists of two modules: (a) Controller identifies learners with valuable parameter update potential, aiming to reduce the cost of parameter generation. (b) Generator adjusts network parameters for existing KT models at different stages or for different groups, aiming to enhance adaptive generalization. In our setup, the KT model is decoupled into a static backbone and a dynamic layer. The generator can be inserted into any layer of the KT model or generate parameters for any layer (dynamic layer). Finally, we introduce the training strategy for Cuff-KT.

### 3.2.1 CONTROLLER

The controller can identify learners with dramatic changes in their knowledge state distribution (*i.e.*, valuable learners, often showing progress or regression), aiming to reduce the cost of parameter generation. The controller comprehensively considers both fine-grained and coarse-grained changes in the knowledge states of different learners, as described below.

**Fine-grained Changes.** At time-step $k$, the KT model models knowledge states States$^j$ (*i.e.*, proficiency scores ranging from 0 to 1 for $|\mathcal{C}|$ concepts over $k$ time steps) for learner $s^j$ with number $j$ ($1 \leq j \leq n$, where $n$ is the total number of learners) at different time steps. The States$^j$ is utilized by the controller to measure the fine-grained distance (*e.g.*, KL-divergence) between the knowledge state distributions across various concept at the intermediate time-step $k/2$ and current time-step $k$:

$$\begin{cases} \text{States}^{*j}_{k/2} = \text{norm}(\text{States}^j_{k/2}), \\ \text{States}^{*j}_k = \text{norm}(\text{States}^j_k), \\ \text{KL}^j = \sum_{c \in \mathcal{C}} \text{States}^{*j}_k(c) \log \dfrac{\text{States}^{*j}_k(c)}{\text{States}^{*j}_{k/2}(c)} + 1, \end{cases} \tag{4}$$

where $\text{norm}(\cdot)$ denotes the normalization operation.

**Coarse-grained Changes** However, focusing solely on fine-grained changes might not capture the overall knowledge state changes of the learners. The Zone of Proximal Development (ZPD) is a core concept in Dynamic Assessment Theory (Vygotsky & Cole, 1978). It refers to the gap between a learner's current independent ability level and the potential level that could be reached with the help of other mediums (*e.g.*, ITS). It describes the overall changes in the learner's knowledge state (*i.e.*, progress or regression). Inspired by this, we consider the overall correct rate at the intermediate time-step $k/2$ as the lower limit of the ZPD, and the correct rate at the current time-step $k$ as the upper limit or near-upper limit of the ZPD. We use the rate of change as a quantitative indicator of the ZPD$^j$ of learner $s^j$:

$$\text{ZPD}^j = \left| \frac{\sum_{i=1}^k r_i^j}{\sum_{i=1}^k 1} - \frac{\sum_{i=1}^{k/2} r_i^j}{\sum_{i=1}^{k/2} 1} \right| \times \text{len}_j \div \left( \frac{\sum_{i=1}^{k/2} r_i^j}{\sum_{i=1}^{k/2} 1} + 1 \right) + 1, \tag{4}$$

where $\text{len}_j$ is the actual length of questions attempted by learner $s^j$ ($\text{len}_j \leq k$, when $\text{len}_j < k$, the missing sequence, *e.g.*, concepts sequence, is often padded with 0), which reflects the reliability of the ZPD$^j$, with a larger $\text{len}_j$ indicating more reliable results.

Finally, the controller assigns a value score to learner $s^j$:

$$\text{score}_j = \text{KL}^j \times \text{ZPO}^j. \tag{5}$$

It can be observed that $\text{KL}^j$ and $\text{ZPO}^j$ are positive, which avoids any absolute impact on the $\text{score}_j$ when either one is 0. Notably, the controller can identify learners who have shown significant progress or regression, which is beneficial for teachers or ITS to pay further attention to them.

### 3.2.2 GENERATOR

The generator can generate personalized dynamic parameters for learners determined by the controller based on real-time samples from different stages or groups, aiming to improve the adaptive generalization for continuously changing distributions. We first introduce the generator's feature extraction, then propose our designed state-adaptive attention, and finally discuss generating parameters through low-rank decomposition. For convenience, we have omitted the superscript of the learner numbers.

**Feature Extraction**. At time-step $k$, the generator takes $\{(c_i, r_i)\}_{i=1}^k$ as input, considering the relative relationship between question difficulty and learner ability (Rasch, 1993) and inspired by the dual-tower model in recommendation systems (Huang et al., 2013; Covington et al., 2016), embedding the questions $c_{1:k}$ and responses $r_{1:k}$ into vector spaces $Q_{1:k} \in \mathbb{R}^d$ and $R_{1:k} \in \mathbb{R}^d$, respectively ($d$ is the dimension of the embedding). After non-linearization, features $H_k^q \in \mathbb{R}^{d_{in}}$ and $H_k^r \in \mathbb{R}^{d_{in}}$ ($d_{in}$ is the input dimension of the dynamic layer) are extracted through a sequential

feature extractor (SFE) (*e.g.*, GRU):

$$\begin{cases} H_k^q = \text{SFE}(\text{Tanh}(Q_{1:k}W_1 + b_1)), \\ H_k^r = \text{SFE}(\text{Tanh}(R_{1:k}W_2 + b_2)), \end{cases} \tag{6}$$

where $W_1 \in \mathbb{R}^{d \times d_{in}}$, $W_2 \in \mathbb{R}^{d \times d_{in}}$, $b_1 \in \mathbb{R}^{d_{in}}$, $b_2 \in \mathbb{R}^{d_{in}}$ are learnable parameters in the projection layer. $\text{Tanh}(\cdot)$ is the activation function.

**State-adaptive Attention (SAA)**. SAA is responsible for adaptive generalization of the extracted question and response features, considering both change in concept correct rate (*i.e.*, difficulty) and the time of the change in knowledge state. Intuitively, the greater the change in difficulty, indicating more significant progress or regression, and the longer the time since the last response, the more likely a sudden change in knowledge state can occur. Such positions should receive more attention. Therefore, the definition of SAA is as follows:

$$\begin{cases} \text{SAA}(X_k) = \text{Concat}(\text{head}_1, \cdots, \text{head}_h)W_h, \\ \text{head}_i = \text{Attention}^*(Q = X_k^{/h}, K = X_k^{/h}, V = X_k^{/h}), \\ \text{Attention}^*(Q, K, V) = \text{softmax}^*(X = \dfrac{QK^T}{\sqrt{d/h}})V, \\ \text{softmax}^*(X) = \text{attn}_w(c_{1:k}, r_{1:k}, t_{1:k}) \cdot \text{softmax}(X), \\ \text{attn}_w(c_{1:k}, r_{1:k}, t_{1:k}) = \text{dist}_d(c_{1:k}, r_{1:k}) \cdot \text{dist}_t(c_{1:k}, t_{1:k}), \end{cases} \tag{5}$$

where $h$ is the number of attention heads and $W_h \in \mathbb{R}^{d_{in} \times d_{in}}$. $X_k^{/h}$ represents splitting the $d_{in}$ dimensions of $X_k$ into $h$ parts. $\text{dist}_d$ and $\text{dist}_t$ represent the changes in difficulty and time, respectively. At position $i \in [1, k]$, $\text{dist}_d(c_i, r_i)$ and $\text{dist}_t(c_i, t_i)$ are respectively:

$$\begin{cases} 1, & \text{if } i = 1, \\ \left( \dfrac{\sum_{j=1}^{i} r_j[c_j = c_i]}{\sum_{j=1}^{i} 1[c_j = c_i]} - \dfrac{\sum_{j=1}^{i-1} r_j[c_j = c_i]}{\sum_{j=1}^{i-1} 1[c_j = c_i]} \right) + 1. & \text{else} \end{cases} \tag{8}$$

$$\begin{cases} 1, & \text{if } j = \max\{k \mid k < i \text{ and } c_k = c_i\} = \emptyset, \\ \dfrac{t_i - t_j}{t_i - t_1}. & \text{else} \end{cases}$$

Finally, the representations $S_k^q$ and $S_k^r$ of question difficulty and learner ability are obtained by SAA:

$$S_k^q = \text{SAA}(H_k^q), S_k^r = \text{SAA}(H_k^r), \tag{9}$$

where $S_k^q$ and $S_k^r$ characterize the difficulty distribution of questions and the ability distribution of learners, respectively, based on real-time data from the current stage or group. SAA is the core component of the generator, and we will further discuss its importance in Sec. 4.5.

**Low-rank Decomposition**. Before performing low-rank decomposition on the parameters, the learned question difficulty $S_k^q$ and learner ability $S_k^r$ are uniformly expressed as the generalized information feature $S_k$ that characterizes the interaction distribution of learners:

$$S_k = S_k^q + S_k^r, \tag{10}$$

Finally, parameters (*i.e.*, weight and bias) are generated through $S_k$ for the dynamic layer:

$$\begin{cases} \text{weight} = S_k W_w + b_w, \\ \text{bias} = S_k W_b + b_b, \end{cases} \tag{11}$$

where $W_w \in \mathbb{R}^{d_{in} \times (d_{in} \times d_{out})}$, $b_w \in \mathbb{R}^{d_{in} \times d_{out}}$, $W_b \in \mathbb{R}^{d_{in} \times d_{out}}$, $b_b \in \mathbb{R}^{d_{out}}$ are learnable parameters. $d_{out}$ is the output dimension of the dynamic layer. However, it can be observed that the parameter size of $W_w$ is too large, which increases computational resources and the risk of overfitting. Inspired by LoRA (Hu et al., 2021), $W_w$ is decomposed into low-rank matrices to obtain the final weight:

$$\text{weight} = S_k W_{w_1} W_{w_2} + b_w, \tag{12}$$

where $W_{w_1} \in \mathbb{R}^{d_{in} \times \text{rank}}$, $W_{w_2} \in \mathbb{R}^{\text{rank} \times (d_{in} \times d_{out})}$ are learnable parameters, and $\text{rank} \ll d_{in}$ is a very small value (*e.g.*, 1). In Sec. 4.5, we will further analyze the effects of different rank.

It's noted that the generator can generate parameters for the dynamic layer, given the input dimension $d_{in}$ and output dimension $d_{out}$. In our experiments, the dynamic layer defaults to the output layer of the KT model.

### 3.2.3 MODEL TRAINING

All learnable parameters are trained by minimizing the binary cross-entropy between $r_i$ and $\hat{r}_i$, *i.e.*,

$$\mathcal{L} = -\sum_{i=1}^{k} r_i \log(\hat{r}_i) + (1 - r_i)\log(1 - \hat{r}_i). \tag{13}$$

## 4 EXPERIMENTS

In this section, we demonstrate the superiority of our proposed Cuff-KT and the impact of its different components through experiments. Specifically, the experimental evaluation is divided into *(i)* the controllability of parameter generation (Sec. 4.2), *(ii)* prediction accuracy, quantifying the effectiveness of tackling RLPA (Sec. 4.3), *(iii)* the application of Cuff-KT (Sec. 4.4), and *(iv)* the impact of dual-tower modeling, SFE, SAA, and low-rank decomposition in Cuff-KT (Sec. 4.5).

### 4.1 EXPERIMENTAL SETUP

#### 4.1.1 DATASETS

We conduct extensive experiments on a classic dataset (assist15 (Feng et al., 2009)) and two recently proposed benchmark datasets (comp (Hu et al., 2023) and xes3g5m (Liu et al., 2024)). Following the data preprocessing method outlined in (Lee et al., 2022b), we exclude learners with fewer than five interactions and all interactions involving nameless concepts. Since a question may involve multiple concepts, we convert the unique combinations of concepts within a single question into a new concept. Table 1 provides a statistical overview of these datasets. It's noted that the large datasets (comp and xes3g5m) are randomly sampled 5000 learners.

See Appendix A.1 for a detailed description of the datasets.

#### 4.1.2 BASELINES

We select a classic KT model (DKT (Piech et al., 2015)) and two recently proposed state-of-the-art models (AT-DKT (Liu et al., 2023a) and DIMKT (Shen et al., 2022)) as the backbone models for

Table 1: Statistics of 3 datasets.

| Datasets | #learners | #questions | #concepts | #interactions |
|----------|-----------|------------|-----------|---------------|
| assist15 | 17,115 | 100 | 100 | 676,288 |
| comp | 5,000 | 7,460 | 445 | 668,927 |
| xes3g5m | 5,000 | 7,242 | 1,221 | 1,771,657 |

optimization. We compare Cuff-KT with these three backbone models and three classic fine-tuning based methods: Full Fine-tuning (FFT), Adapter-based tuning (Adapter) (Houlsby et al., 2019), and Bias-term Fine-tuning (BitFit) (Zaken et al., 2021).

See detailed introductions to the backbone models and baselines in Appendix A.3.

#### 4.1.3 IMPLEMENTATION

We implement all models using Pytorch on a Linux server with two GeForce RTX 3090s. We used the Adam optimizer with a learning rate of 0.001, and a batch size of 512. The embedding dimension for all models is fixed at 32. The rank of the generator in Cuff-KT is set to 1. We split the historical interactions into training, validation, and test sets (7:2:1) based on timestamps and groups, respectively. An early stopping strategy is applied if the AUC on the validation set does not increase for 10 epochs. The experiments are repeated 5 times under random seeds 0 to 4 and the average performance is reported. Following the previous works (Piech et al., 2015; Shen et al., 2022; Liu et al., 2023a),the evaluation metrics include Area Under the ROC Curve (AUC) and Root Mean Square Error (RMSE).

### 4.2 CONTROLLABLE PARAMETER GENERATION

According to (Lv et al., 2023a), anomaly detection algorithms can be used to detect distribution changes over time. We select four representative anomaly detection algorithms from pyod library (Zhao et al., 2019) as comparison baselines for the controller in Cuff-KT: LOF (Breunig et al.,

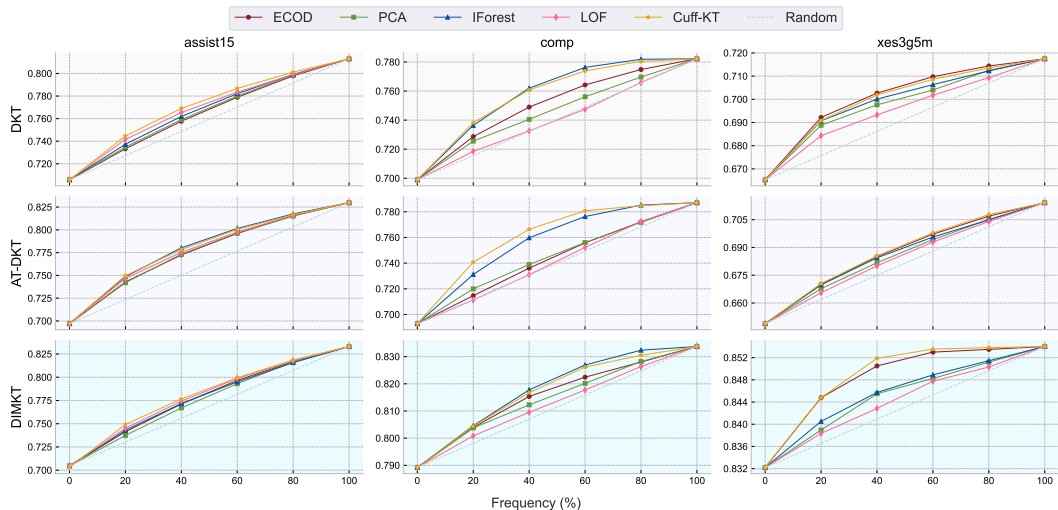

Figure 4: Performance comparison of Cuff-KT and anomaly detection algorithms at different frequencies.

Table 2: Performance comparison between different methods under intra-learner shift. The **best result** is in bold and the next best is underlined. * and ** indicate that the improvements over the strongest baseline are statistically significant, with $p < 0.05$ and $p < 0.01$, respectively.

| Dataset→ | assist15 | | | comp | | | xes3g5m | | |
|---|---|---|---|---|---|---|---|---|---|
| Method↓\Metric→ | AUC ↑ | RMSE ↓ | Time Cost ↓ | AUC | RMSE | Time Cost | AUC | RMSE | Time Cost |
| DKT | 0.7058 | 0.4107 | 0ms | 0.6990 | 0.3613 | 0ms | 0.6633 | 0.4129 | 0ms |
| +FFT | 0.7063 | 0.4071 | ≥17,200ms | 0.7066 | 0.3594 | ≥18,300ms | 0.7116 | 0.3992 | ≥33,600ms |
| +Adapter | 0.6749 | 0.4242 | ≥16,600ms | 0.6634 | 0.3714 | ≥17,700ms | 0.6467 | 0.4275 | ≥36,100ms |
| +BitFit | 0.7054 | 0.4080 | ≥16,300ms | 0.7039 | 0.3599 | ≥14,100ms | 0.6841 | 0.4105 | ≥32,600ms |
| **+Cuff-KT** | **0.8130**** | **0.3773**** | ≥419ms | **0.7834**** | **0.3459**** | ≥435ms | **0.7176** | **0.3931** | ≥1,211ms |
| AT-DKT | 0.6981 | 0.4106 | 0ms | 0.6922 | 0.3621 | 0ms | 0.6437 | 0.4228 | 0ms |
| +FFT | 0.7005 | 0.4083 | ≥126,400ms | 0.7020 | 0.3602 | ≥95,000ms | 0.6918 | 0.4068 | ≥176,000ms |
| +Adapter | 0.6588 | 0.4287 | ≥125,100ms | 0.6443 | 0.3878 | ≥88,800ms | 0.6276 | 0.4351 | ≥168,300ms |
| +BitFit | 0.6989 | 0.4094 | ≥121,300ms | 0.6990 | 0.3608 | ≥91,300ms | 0.6668 | 0.4178 | ≥169,300ms |
| **+Cuff-KT** | **0.8335**** | **0.3714**** | ≥236ms | **0.7869**** | **0.3435**** | ≥254ms | **0.7133**** | **0.4009*** | ≥784ms |
| DIMKT | 0.7055 | 0.4080 | 0ms | 0.7934 | 0.3404 | 0ms | 0.8322 | 0.3402 | 0ms |
| +FFT | 0.7072 | 0.4063 | ≥270,900ms | 0.8000 | 0.3375 | ≥205,200ms | 0.8366 | 0.3383 | ≥377,800ms |
| +Adapter | 0.6507 | 0.4387 | ≥410,000ms | 0.7526 | 0.3671 | ≥278,500ms | 0.7929 | 0.3696 | ≥509,200ms |
| +BitFit | 0.7082 | 0.4061 | ≥263,500ms | 0.7972 | 0.3382 | ≥199,400ms | 0.8369 | 0.3381 | ≥347,800ms |
| **+Cuff-KT** | **0.8322**** | **0.3710*** | ≥232ms | **0.8380**** | **0.3297**** | ≥175ms | **0.8540*** | **0.3347*** | ≥239ms |

2000), PCA (Shyu et al., 2003), IForest (Liu et al., 2008), and ECOD (Li et al., 2022), and use AUC as the evaluation metric. Detailed descriptions of these four algorithms can be found in the Appendix A.2. Figure 4 shows the performance results under intra-learner shift when the controller selects learners with different frequencies for the generator.

We can see that anomaly detection algorithms (especially IForest and ECOD) consistently outperform the random selection, demonstrating the correctness of using anomaly detection algorithms to detect distribution changes. Moreover, our Cuff-KT generally performs better than these algorithms, indicating that Cuff-KT is more capable of identifying learners whose model generalization deteriorates due to distribution changes. We attribute Cuff-KT's breakthrough to the Dynamic Assessment Theory (Vygotsky & Cole, 1978), which we further analyze in the Appendix A.4.

### 4.3 TUNING-FREE AND FAST PREDICTION

Under this setting, the generator in Cuff-KT generates parameters for all learners independently of the controller. In our setup, we attempt to divide learners into different groups based on the degree of change in their knowledge states. We use DKT to encode each learner's interaction history and choose the distance (*e.g.*, KL divergence) between the prediction distributions for each concept at the

Table 3: Performance comparison between different methods under inter-learner shift. The **best result** is in bold and the next best is underlined. * and ** indicate that the improvements over the strongest baseline are statistically significant, with $p < 0.05$ and $p < 0.01$, respectively.

| Dataset→ | assist15 | | | comp | | | xes3g5m | | |
|---|---|---|---|---|---|---|---|---|---|
| Method↓\Metric→ | AUC ↑ | RMSE ↓ | Time Cost ↓ | AUC | RMSE | Time Cost | AUC | RMSE | Time Cost |
| DKT | 0.7075 | 0.4363 | 0ms | 0.6681 | 0.4355 | 0ms | 0.7907 | 0.4329 | 0ms |
| +FFT | 0.7137 | 0.4339 | ≥18,800ms | 0.6839 | 0.4310 | ≥3,600ms | 0.7990 | 0.4166 | ≥4,400ms |
| +Adapter | 0.6805 | 0.4456 | ≥17,000ms | 0.6461 | 0.4438 | ≥3,200ms | 0.7646 | 0.4427 | ≥4,300ms |
| +BitFit | 0.7119 | 0.4349 | ≥17,200ms | 0.6734 | 0.4326 | ≥3,100ms | 0.7905 | 0.4323 | ≥4,900ms |
| **+Cuff-KT** | **0.7365*** | **0.4302** | ≥355ms | **0.6937**** | **0.4294*** | ≥96ms | **0.8004** | **0.4158** | ≥123ms |
| AT-DKT | 0.7030 | 0.4389 | 0ms | 0.6587 | 0.4375 | 0ms | 0.7868 | 0.4370 | 0ms |
| +FFT | 0.7104 | 0.4355 | ≥74,700ms | 0.6751 | 0.4312 | ≥20,300ms | 0.7916 | 0.4242 | ≥21,300ms |
| +Adapter | 0.6708 | 0.4520 | ≥55,300ms | 0.6253 | 0.4498 | ≥18,100ms | 0.7643 | 0.4457 | ≥23,500ms |
| +BitFit | 0.7076 | 0.4367 | ≥59,100ms | 0.6666 | 0.4334 | ≥18,600ms | 0.7860 | 0.4352 | ≥19,300ms |
| **+Cuff-KT** | **0.7348**** | **0.4316*** | ≥170ms | **0.6919**** | **0.4303*** | ≥64ms | **0.7959*** | **0.4183*** | ≥110ms |
| DIMKT | 0.7134 | 0.4350 | 0ms | 0.7556 | 0.4118 | 0ms | 0.8255 | 0.4088 | 0ms |
| +FFT | 0.7187 | 0.4320 | ≥173,500ms | 0.7590 | 0.4097 | ≥52,500ms | **0.8329** | **0.3983** | ≥48,200ms |
| +Adapter | 0.6648 | 0.4577 | ≥217,100ms | 0.7017 | 0.4465 | ≥82,500ms | 0.7467 | 0.4618 | ≥81,800ms |
| +BitFit | 0.7144 | 0.4334 | ≥154,400ms | 0.7563 | 0.4110 | ≥50,800ms | 0.8254 | 0.4084 | ≥48,600ms |
| **+Cuff-KT** | **0.7425**** | **0.4296** | ≥203ms | **0.7657**** | **0.4057*** | ≥64ms | 0.8309 | 0.4009 | ≥72ms |

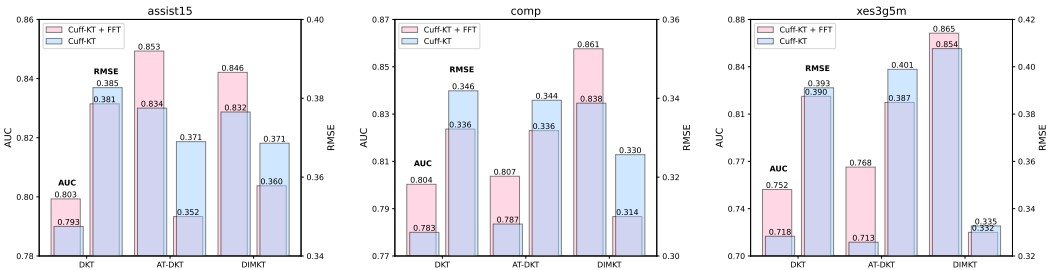

Figure 5: Cuff-KT+FFT under intra-learner shift.

intermediate and current timestamps as the basis for division. Tables 2 and 3 show the performance comparison between different methods under intra-learner shift and inter-learner shift. Overall, our Cuff-KT effectively tackles the RLPA task with significant advantages. We can observe:

- Overall, compared to baseline methods, our Cuff-KT generally performs best on all metrics across all datasets. This performance improvement can be attributed to Cuff-KT's parameter generation approach, which dynamically updates the model to capture distribution dynamics rather than statically considering interactions in the test data, enhancing the KT model's dynamic adaptability.

- Compared to the backbone, the time cost caused by Cuff-KT is significantly smaller than fine-tuning-based methods. This is because Cuff-KT updates model parameters only through feedforward computation, without the need for a retraining process.

- Adapter fine-tuning performs poorly and even leads to performance degradation, as it is heavily affected by task complexity and model scale (He et al., 2021; Karimi Mahabadi et al., 2021), ultimately resulting in overfitting.

- Although FFT and BitFit fine-tuning methods generally improve the performance of the backbone, especially FFT based on DKT showing a 0.483 increase in AUC metric on the xes3g5m dataset under intra-learner shift, the time cost caused is non-negligible in real-world scenarios.

## 4.4 FLEXIBLE APPLICATION

Thanks to the independence of the generator in our Cuff-KT from fine-tuning based methods, we attempt to combine Cuff-KT with FFT. The results in terms of AUC and RMSE under intra-learner shift and inter-learner shift are shown in Figure 5 and Figure 8 in the Appendix A.4, respectively. As can be seen from the figures, on different backbone models and across all datasets, the performance still shows a significant improvement after combining Cuff-KT with FFT. This is because FFT can learn different distributions from the recent data, facilitating Cuff-KT's smooth transition to the

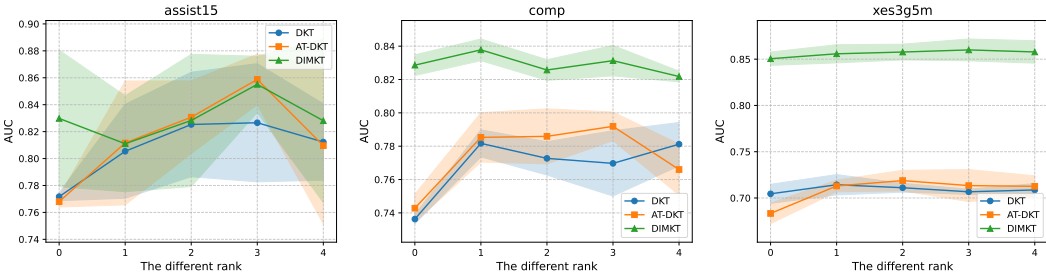

Figure 6: Performance on AUC for different rank under intra-learner shift.

distribution in the test data. This combination provides a reference for flexibly fine-tuning models in special real-world scenarios where real-time requirements are not high.

Moreover, the generator in Cuff-KT can flexibly generate parameters for or insert into any layer of the KT model. This inspires us to consider how the generator can choose the position and network structure for generation or insertion. Due to space limitations, we leave this as a direction for future research.

### 4.5 ABLATION STUDY

We systematically examine the impact of key components in Cuff-KT based on DKT by constructing four variants under intra-learner shift. "**w/o.** Dual" indicates that question and response embeddings are fused (*e.g.*, by summation) after embedding. "**w/o.** SFE" means the SFE component is omitted, "**w/o.** SAA" means omitting the SAA component, and "**w.** SHA" means SAA is replaced with standard multi-head attention. From Table 4,

Table 4: The performance of different variants in Cuff-KT.

| Dataset→ | assist15 | | comp | | xes3g5m | |
|---|---|---|---|---|---|---|
| Metric→ | AUC↑ | RMSE↓ | AUC | RMSE | AUC | RMSE |
| **Cuff-KT** | **0.8130** | **0.3773** | **0.7834** | **0.3459** | **0.7176** | **0.3931** |
| **w/o.** Dual | 0.7013 | 0.4126 | 0.7245 | 0.3693 | 0.7088 | 0.4094 |
| **w/o.** SFE | 0.7706 | 0.3925 | 0.7204 | 0.3612 | 0.6896 | 0.4140 |
| **w/o.** SAA | 0.7000 | 0.4141 | 0.6877 | 0.3640 | 0.6716 | 0.4212 |
| **w.** SHA | 0.7810 | 0.3844 | 0.6924 | 0.3629 | 0.6767 | 0.4185 |

we can easily observe: (1) Cuff-KT outperforms all variants, especially when the SAA component is removed, the predictive performance generally decreases the most, while Cuff-KT with standard multi-head attention comes next, empirically validating that our designed SAA component can effectively achieve adaptive generalization. (2) Cuff-KT's performance is very low when the SFE component is removed or dual modeling is not employed. We believe this is because Cuff-KT can successfully extract question features and learner response features and effectively learn the difficulty distribution of current questions and the ability distribution of learners based on real-time data.

Additionally, we study the effects of different rank under intra-learner shift. The performance on AUC of different rank under intra-learner shift and the parameter size of the generator in Cuff-KT are shown in Figure 6 and Table 5 in the Appendix A.4, respectively. In Figure 6, after low-rank decomposition (rank $\neq$ 0), the performance on AUC generally improves, and the effects of different rank are inconsistent across different datasets. In Table 5, the parameter size of the generator increases with the rank, indicating that by adjusting different ranks, an effective balance between the performance and resource consumption of Cuff-KT can be achieved.

## 5 CONCLUSION

Our paper aims to tackle the RLPA task in KT by proposing a controllable, tuning-free, fast, and flexible method called Cuff-KT to improve adaptability of KT models in real-world scenarios. We decompose the RLPA task to be solved into two sub-issues: intra-learner shift and inter-learner shift, and design a parameter generator capable of generate personalized parameters based on the current stage or group, thereby achieving adaptive generalization. In instance validations across multiple KT models, Cuff-KT exhibits superior performance in adapting to rapidly changing distributions, avoiding the overfitting and high time cost challenges inherent in fine-tuning based methods.

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

# A  APPENDIX

In this document, we include the description of the datasets (A.1), an overview to anomaly detection algorithms (A.2), an introduction to the backbone models and baselines ( A.3), and additional experimental results (A.4), which we are unable to include in the main paper due to space limitations.

## A.1  DATASETS

Here, we describe the datasets (assist15, comp, xes3g5m) used for evaluation.

- **assist15**[3] (Feng et al., 2009): The assist15 dataset, is collected from the ASSISTments platform in the year of 2015. It includes a total of 708,631 interactions involving 100 distinct concepts from 19,917 learners.

- **comp**[4] (Hu et al., 2023): The comp dataset, is part of the PTADisc, which encompasses a wide range of courses from the PTA platform. PTADisc includes data from 74 courses, involving 1,530,100 learners and featuring 4,504 concepts, 225,615 questions, as well as an extensive log of over 680 million learner responses. The comp dataset is specifically selected for KT task in Computational Thinking course.

- **xes3g5m**[5] (Liu et al., 2024): The xes3g5m dataset incorporates rich auxiliary information such as tree-structured concept relationships, question types, textual contents, and learner response timestamps and includes 7,652 questions and 865 concepts, with a total of 5,549,635 interactions from 18,066 learners.

## A.2  ANOMALY DETECTION ALGORITHMS

We compare the controller of Cuff-KT with the following anomaly detection algorithms:

- **LOF** (Breunig et al., 2000): LOF quantifies the local outlier degree of samples by calculating a score. This score reflects the ratio of the average density of the local neighborhood around a sample point to the density at the location of that sample point. A ratio significantly greater than 1 indicates that the density at the sample point's location is much lower than the average density of its surrounding neighborhood, suggesting that the point is more likely to be a local outlier.

- **PCA** (Shyu et al., 2003): After performing eigenvalue decomposition, the eigenvectors obtained from PCA reflect different directions of variance change in network traffic data, while eigenvalues represent the magnitude of variance in the corresponding directions. Thus, the eigenvector associated with the largest eigenvalue represents the direction of maximum variance in network traffic data, while the eigenvector associated with the smallest eigenvalue represents the direction of minimum variance. If an individual network connection sample exhibits characteristics inconsistent with the overall network traffic sample, such as deviating significantly from other normal connection samples in certain directions, it may indicate that this connection sample is an outlier.

- **IForest** (Liu et al., 2008): IForest employs an innovative anomaly isolation method to identify anomalous samples by constructing a binary tree structure (called an Isolation Tree or iTree). Unlike traditional methods, IForest does not build a model of normal samples, but instead directly isolates anomalies. In this process, anomalous samples tend to be isolated more quickly and thus are positioned closer to the root node in the tree, while normal samples are isolated deeper in the tree. By constructing multiple iTrees (typically T trees), the average path length from anomalies to the root node is significantly shorter than that of normal points, and this characteristic is used for anomaly detection. This approach excels in handling large-scale datasets and high-dimensional data, with the advantages of linear time complexity and low memory requirements.

---

[3]https://sites.google.com/site/assistmentsdata/datasets/
2015-assistments-skill-builder-data
[4]https://github.com/wahr0411/PTADisc
[5]https://github.com/ai4ed/XES3G5M

- **ECOD** (Li et al., 2022): ECOD is a novel unsupervised anomaly detection algorithm. Its core idea stems from the definition of outliers—typically rare events occurring in the tails of a distribution. The algorithm cleverly uses empirical cumulative distribution functions (ECDF) to estimate the joint cumulative distribution function of the data, thereby calculating the probability of outliers. The uniqueness of ECOD lies in its avoidance of the slow convergence problem of joint ECDF in high-dimensional data. The algorithm calculates the univariate ECDF for each dimension separately, then estimates the degree of anomaly for multidimensional data points through an independence assumption. This is done by multiplying the estimated tail probabilities of all dimensions.

## A.3 BACKBONE MODELS AND BASELINES

We instantiate a classic backbone model and two recently proposed SOTA models.

- **DKT** (Piech et al., 2015): DKT is a seminal model that leverages Recurrent Neural Networks (RNNs), specifically utilizing a single layer LSTM, to directly model learners' learning processes and predict their performance.
- **AT-DKT** (Liu et al., 2023a): AT-DKT augments the original deep knowledge tracing model by embedding two auxiliary learning tasks: one for predicting concepts and another for assessing individualized prior knowledge. This integration aims to sharpen the model's predictive accuracy and deepen its understanding of learner performance.
- **DIMKT** (Shen et al., 2022): DIMKT is designed to enhance the assessment of learners' knowledge states by explicitly incorporating the difficulty level of questions and establishes the relationship between learners' knowledge states and difficulty level during the practice process.

We compare Cuff-KT with three classic fine-tuning based methods.

- **Full Fine-tuning (FFT)**: FFT involves training all parameters of a model completely. It usually has the highest potential for performance, but it consumes the most resources, takes the longest time to train, and is prone to overfitting when the corpus is not large enough.
- **Adapter-based tuning (Adapter)** (Houlsby et al., 2019): Adapter inserts downstream task parameters, known as adapters, into each Transformer block of the pre-trained model. Each adapter consists of two layers of MLP and an activation function, responsible for reducing and increasing the dimensionality of the Transformer's representations. During fine-tuning, the main model parameters are frozen, and only the task-specific parameters are trained. Since the backbone models might not include a Transformer, in our experiments, it is replaced by linear layers.
- **Bias-term Fine-tuning (BitFit)** (Zaken et al., 2021): BitFit is a sparse fine-tuning method that efficiently tunes only the parameters with bias, while all other parameters are fixed. This method tends to be effective on small to medium datasets and can even compete with other sparse fine-tuning methods on large datasets.

## A.4 ADDITIONAL EXPERIMENTAL RESULTS

1. We further analyze the influence of different components of the controller in Cuff-KT under intra-learner shift. We instantiate DKT on assist15, comp, and xes3g5m datasets. The AUC performance results are shown in Figure 7. We observe that the performance drops the most when the controller removes ZPD ("w/o. ZPD", *i.e.*, without considering coarse-grained changes in knowledge states). This indicates that considering coarse-grained knowledge state changes is crucial, which aligns with reality, as in practical scenarios, a learner's progress or regression is often judged by an overall score. Additionally, when ZPD does not take into account actual length ("w/o. Rel.", *i.e.*, without considering the reliability of ZPD), the performance drops the second most. This is because when a learner has more activity records, their knowledge state is more likely to experience drastic changes, and such learners should receive more attention. On the other hand, when a learner has limited records, the simulated changes in their knowledge state are less reliable and should be given lower weight. When the controller does not consider fine-grained changes in the knowledge

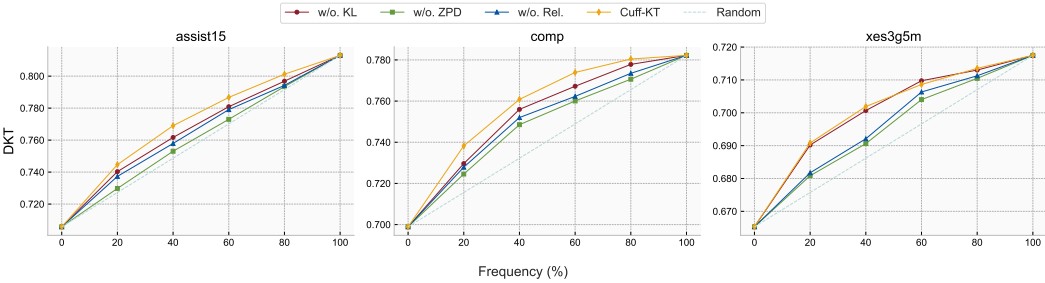

Figure 7: Ablation study of the controller in Cuff-KT at different frequencies.

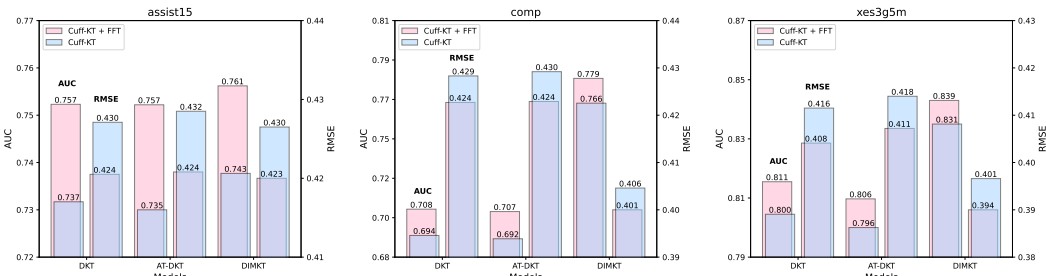

Figure 8: Cuff-KT+FFT under inter-learner shift.

state ("w/o. KL"), the performance shows a slight decrease. We attribute this to the fact that when fine-grained knowledge states decline overall, the learner's knowledge state will experience a major shift. However, such situations are relatively rare in reality.

2. Figure 8 shows the performance of Cuff-KT combined with FFT under inter-learner shift in terms of AUC and RMSE.

3. Table 5 shows the parameter sizes (k) of the generator in Cuff-KT with different ranks under intra-learner shift.

Table 5: The parameter size (k) of the generator with different rank in Cuff-KT under intra-learner shift.

| Dataset | Backbone | Rank | | | | |
|---------|----------|------|------|------|------|------|
| | | 0 | 1 | 2 | 3 | 4 |
| assist15 | DKT | 130.12 | 29.96 | 33.22 | 36.49 | 39.75 |
| | AT-DKT | 130.12 | 29.96 | 33.22 | 36.49 | 39.75 |
| | DIMKT | 54.98 | 23.26 | 24.32 | 25.38 | 26.43 |
| comp | DKT | 516.86 | 74.46 | 88.77 | 103.07 | 117.37 |
| | AT-DKT | 516.86 | 74.46 | 88.77 | 103.07 | 117.37 |
| | DIMKT | 66.02 | 34.30 | 35.36 | 36.42 | 37.47 |
| xes3g5m | DKT | 1,386.76 | 174.57 | 213.70 | 252.84 | 291.97 |
| | AT-DKT | 1,386.76 | 174.57 | 213.70 | 252.84 | 291.97 |
| | DIMKT | 90.85 | 59.14 | 60.19 | 61.25 | 62.30 |

