# OpenReview forum: "Cuff-KT: Tackling Learners' Real-time Learning Pattern Adjustment via Tuning-Free Knowledge State-Guided Model Updating"
_ICLR.cc/2025/Conference — ICLR 2025 Conference Withdrawn Submission_

### Official Review · Reviewer_Y6Gp · 2024-10-22

[review text omitted: it was posted to a different submission]

---

### Official Review · Reviewer_o9EB · 2024-10-30

**Soundness:** 4
**Presentation:** 3
**Contribution:** 3
**Rating:** 5
**Confidence:** 4

**Summary:**

This paper presents the CUFF-KT, which tackles real-time learning pattern adjustment in knowledge tracing by leveraging a tuning-free knowledge state-guided model update approach. CUFF-KT comprises controller and generator modules, capable of generating personalized parameters to adapt quickly to changes in learner distributions. Experiments demonstrate that CUFF-KT significantly improves model performance under various learning pattern shifts.

**Strengths:**

1. This paper use a motivation study to identify a new task called Real-time Learning Pattern Adjustment  which aims at capturing the dynamic nature of different learners’ evolving learning patterns.
2. The proposed Cuff-KT appears to address the issues effectively compared with the selected baselines, as indicated by the experimental results.
3. The structure and framework figure of the paper is clear and easy to follow, making it accessible for readers. However, some formulas are somewhat complex and may benefit from further clarification.

**Weaknesses:**

1. **My biggest concern is why the paper introduces the topic from the perspective of distribution shift and compares with anomaly detection methods.**  How are these two concepts related? My understanding is that it’s one of the causes of distribution shift. The term "anomaly detection" first appears in line 377. What constitutes an anomaly in knowledge tracing? After reading the motivation study, I speculate that anomaly detection in knowledge tracing might refer to changes in students' accuracy rates. In summary, the relationship of two concepts could benefit from more explanation and clarification in the introduction and related work.
2. If this is indeed anomaly detection, then I believe the authors should compare their method with HD-KT[1] to validate its effectiveness.
3. The author cited numerous papers in the related work, such as iAKT [2], which address handling different distributions. Why weren’t these methods included for comparison?

[1] Ma, Haiping, et al. "HD-KT: Advancing Robust Knowledge Tracing via Anomalous Learning Interaction Detection." Proceedings of the ACM on Web Conference 2024. 2024.

[2] Wong, Cheryl Sze Yin, et al. "Incremental context aware attentive knowledge tracing." ICASSP 2022

**Questions:**

1. **Clarification on Distribution Shift and Anomaly Detection**: The paper introduces distribution shift and compares it to anomaly detection methods, but the relationship between these concepts isn’t fully explained. Further clarification is needed on how anomaly detection contributes to or causes distribution shift, especially since anomaly detection first appears in line 377. A clearer explanation in the introduction and related work would help readers understand how "anomaly" is defined within the context of knowledge tracing.

2. **Comparison with HD-KT**: If the method is indeed related to anomaly detection, a comparison with HD-KT, which also addresses anomalous learning interactions, would strengthen the validation of the proposed method.

3. **Comparison with Distribution-Handling Methods like iAKT**: While the authors reference several methods in the related work section that handle different distributions (e.g., iAKT), these methods are not included in the experimental comparisons. A comparison with these approaches would provide a more comprehensive evaluation of the method's effectiveness.

---

### Official Review · Reviewer_BACB · 2024-10-31

**Soundness:** 2
**Presentation:** 3
**Contribution:** 2
**Rating:** 5
**Confidence:** 4

**Summary:**

This work proposes a novel method  called Controllable, tUning- free, Fast, and Flexible Knowledge Tracing (Cuff-KT) to address intra- and inter-learner shifts in student performance prediction. Cuff-KT has two components, a controller and generator mechanism, to learn a model parameter generator specific to the current stage or group and generate updating personalized parameters for valuable learners (e.g., those showing significant progress or regression).

**Strengths:**

I acknowledge the problem setting in this paper is unexplored, regarding the performance distribution shift in educational datasets coming from both students' progress and the online platform's selection mechanism.
The authors also effectively present the problem and motivation, offering a clear and accessible read. I appreciate that Cuff-KT shows a first step to bring this problem into the public, instead of only relying on heavy neural networks and large data to chase the SOTA performance.

Regarding the method, including the controller and the generator, which I also find them very interesting. Though the details of each part are a bit confusing to me (see weaknesses), figure 3 effectively visualizes the overall Cuff-KT pipeline

**Weaknesses:**

The literature review omits recent models like QIKT, SimpleKT, and PSIKT. Including these would clarify Cuff-KT’s positioning. Additionally, the rationale for selecting the three evaluation baselines needs more explanation.

The clarifications of the experiment setup and details of the methodology need improvement, but do correct me if I am missing anything here:

1.	In lines 226 and 431, the authors mention “e.g., KL-divergence,” but it is unclear if other distance measures were considered. It would also be helpful to know how the distribution over embeddings was approximated—was it, for instance, fitted to a Gaussian?
2.	In the controller setting, is there any specific reason that the authors chose to compare step $k$ and intermediate time-step $\frac{k}{2}$, i.e., why not using a sliding window setting? Theoretically, as the time step $k$ goes up, the discrepency between the two distributions will not decrease. This means the $\mathrm{KL}^j$ in Eq.4 will always become larger. Does the model prioritize students with longer learning histories, thereby increasing their likelihood of selection?
3.	In Sec. 3.2.2, I would need much more clarifications so that I can comment on this part. In line 265, $c_i$ and $r_i$ are input for question difficulty and learner ability. Why do authors choose $c_i$ as the input or it is a typo? This question applies to all of the question/concept notations in this section, e.g., “embedding the questions $c_{1: k}$”. I will try to comment on the soundness of generator again after the authors clarify this section. Meanwhile, how do authors embed the time information, i.e., what is $t_{1:k}$ in Eq. 5?
4.	The experiment training and validation needs more clarifications. In lines 368-370, the authors say the historical interactions are split into training, validation, and test sets based on the timestamps and groups. I am not clear how inter- and intra- evaluations are done and how the training and test work, could the authors provide better explanations (more details can come in the appendix if it exceeds page limit)?
5.	Do the authors provide the effectiveness of the controller? How do the performance change if you remove the controller at all, i.e., the model will generate the parameters for every student? I also wonder the statistics of the student chosen by the controller, how many students are chosen in general and what is the correlation between the chosen students and their interaction length? As I wonder in question 2, if the length plays a dominant role (as the authors show in Fig. 7), what else is determining the chosen probability intuitively?

**Questions:**

I would appreciate the authors could answer my questions and add additional experiments/clarifications in the manuscript I proposed in the weakness.

I need further clarification on the remaining questions:

1.	In Eq.2, is the time index $u$ not the actual timestamp but rather the $u$-th interaction of  student $s$. Why did the authors choose to compare students based on their individual learning progress instead of at an arbitrary time horizon? Was this choice made for engineering or implementation convenience?
2.	In Sec. 4.2, I appreciate the inclusion of anomaly detection. However, where does the ground truth for anomaly points in the student KT datasets come from? Do these datasets provide these points?
3.	I suggest the authors revise the presentation of Figures 5 and 8. The overlapping purple colors create confusion; a standard bar plot would improve clarity significantly.

---

### Official Review · Reviewer_34FU · 2024-11-04

**Soundness:** 3
**Presentation:** 2
**Contribution:** 2
**Rating:** 3
**Confidence:** 5

**Summary:**

This paper introduces the concept of Real-time Learning Pattern Adjustment (RLPA), which highlights the intra-learner and inter-learner shifts that affect learners’ knowledge states due to factors like cognitive fatigue and external stress. To address the proposed challenge, the authors propose a novel method called Cuff-KT, which consists of a controller that identifies learners with dramatic changes in their knowledge state distribution and a generator that produces personalized parameters without requiring retraining. This approach allows for fast and flexible adaptation to distribution changes, effectively improving KT models' performance.

**Strengths:**

The study has shown improvement on different KT base models and some finetune methods on a wide range of datasets. Furthermore, several figures are drawn in this paper with bright colors and clear layouts.

**Weaknesses:**

1.	Some mistakes regarding presentation:

a.	In Line 251 $ZPO^j$ should be $ZPD^j$ (The same is true in the source code `train.py`);

b.	In Line 328, $k$ indicates the number of samples, but it has been defined in Line 265 (time-step $k$);

c.	The use of notation in the paper is confusing, lacking a clear distinction between matrices, vectors and scalars, e.g., Line 274 and Line 313.

d.	The counter of equations does not work well. For example, there are two Eq. (4) in page 5.

e.	The citation format is inconsistent; for example, the conference paper lacks the address.

2.	The task RLPA seems like the specification of the distribution shift in time series forecasting in knowledge tracing scenarios [1]. It’s believed that temporal distribution shifts are quite common in time series forecasting and has many solutions. However, the paper does not mention anyone in related work or make any of them compared with the proposed method.

3.	Line 204 is unrelated to the proposed RLPA task. Minimizing this KL divergence is the same as the BCE loss in Line 328, which is the typical KT task loss. This formula doesn't hold any practical significance here.

4.	The paper doesn’t clearly explain the experimental setup. For example, it’s hard to find a straightforward description of how datasets are divided in settings of inter-learner and intra-learner respectively; you have to look at the code to figure it out.

References:

[1] Fan et al. Dish-TS: A General Paradigm for Alleviating Distribution Shift in Time Series Forecasting. In _Proceedings of the 37th AAAI Conference on Artificial Intelligence_, Washington, DC.

**Questions:**

In addition to the previously mentioned weaknesses, there are some other questions:

1.	Is it really necessary to fine-tune to solve this problem? In addition to the framework mentioned, it should also compare their framework with some SOTA KT methods to see if their approach improves upon the basic methods more than the SOTA methods do.

2.	Why does the source code include the LoRA method, yet this is not used as a baseline in experiments?

3.	Why is the AUC on xes3g5m in the setting of intra-learner less than that in the inter-learner? Since AUC on assist15 and comp in the setting of intra-learner is greater than that in the inter-learner.

4.	In Line 244, what is the rationale for choosing $k/2$? Why isn't the $\sum_{i=1}^k1$ simply written as $k$ here?

5.	A more detailed explanation of the specific educational implications of the formulas in lines 293-298 could be provided.

I will raise my score if these questions are addressed.

---

### Note · Authors · 2024-11-17

I have read and agree with the venue's withdrawal policy on behalf of myself and my co-authors.